# Investigating the Triple Code Model in numerical cognition using stereotactic electroencephalography

Alexander P. Rockhill[1], Hao Tan[1], Christian G. Lopez Ramos[1], Caleb Nerison[2], Beck Shafie[1], Maryam N. Shahin[1], Adeline Fecker[1], Mostafa Ismail[1], Daniel R. Cleary[1], Kelly L. Collins[1], Ahmed M. Raslan[1] *

1 Departments of Neurological Surgery, Oregon Health & Science University, Portland, Oregon, United States of America, 2 Department of Family Medicine, Lexington Medical Center, West Columbia, South Carolina, United States of America

* raslana@ohsu.edu

**Data Availability Statement:** The dataset used in this study is available on https://openneuro.org/datasets/ds005415.

## Abstract

The ability to conceptualize numerical quantities is an essential human trait. According to the "Triple Code Model" in numerical cognition, distinct neural substrates encode the processing of visual, auditory, and non-symbolic numerical representations. While our contemporary understanding of human number cognition has benefited greatly from advances in clinical imaging, limited studies have investigated the intracranial electrophysiological correlates of number processing. In this study, 13 subjects undergoing stereotactic electroencephalography for epilepsy participated in a number recognition task. Drawing upon postulates of the Triple Code Model, we presented subjects with numerical stimuli varying in representation type (symbolic vs. non-symbolic) and mode of stimuli delivery (visual vs. auditory). Time-frequency spectrograms were dimensionally reduced with principal component analysis and passed into a linear support vector machine classification algorithm to identify regions associated with number perception compared to inter-trial periods. Across representation formats, the highest classification accuracy was observed in the bilateral parietal lobes. Auditory (spoken and beeps) and visual (Arabic) number formats preferentially engaged the superior temporal cortices and the frontoparietal regions, respectively. The left parietal cortex was found to have the highest classification for number dots. Notably, the putamen exhibited robust classification accuracies in response to numerical stimuli. Analyses of spectral feature maps revealed that non-gamma frequency, below 30 Hz, had greater-than-chance classification value and could be potentially used to characterize format specific number representations. Taken together, our findings obtained from intracranial recordings provide further support and expand on the Triple Code Model for numerical cognition.

**Funding:** The author(s) received no specific funding for this work.

**Competing interests:** The authors have declared that no competing interests exist.

## Introduction

The capacity to comprehend numerical quantities is remarkably phylogenetically conserved across several animal species [1]. However, the ability to deftly conceptualize and leverage abstract symbolic representations of numerical quantities is a uniquely human trait [1, 2]. Given the ubiquitous need to interact with quantities in daily life and the deleterious quality-of-life consequences associated with poor number literacy [3], developing further insights into the neural basis for our sophisticated numerical processing capabilities is of particular interest and value.

A leading framework for numerical cognition is the "Triple Code Model" (TCM) [4, 5]. The TCM postulates that cortical neuronal populations within the ventral temporal lobe, perisylvian region, and intraparietal sulcus are responsible for processing visual ("6"), auditory verbal ("six"), and non-symbolic ("•••••") numerical representation formats, respectively [5]. Studies utilizing functional magnetic resonance imaging (fMRI) and transcranial magnetic stimulation (TMS) have lent support to the TCM while also indicating a broader engagement of cortical regions beyond those originally proposed, such as the cingulate gyrus and cerebellum [6, 7]. Invasive modalities such as intracranial electroencephalography have also affirmed the TCM, contributing to a more robust understanding of the neuronal spatiotemporal dynamics during number processing [8–11].

To date, iEEG investigations have primarily relied on electrocorticography (ECoG) in the form of subdural strips and grids to analyze cortical areas. Despite compelling evidence that subcortical structures play an integral role in number processing [12–15], exceptionally few studies have conducted electrophysiological recordings from deeper subcortical regions [15, 16]. To address this gap, we used stereotactic electroencephalography (sEEG) recordings in medically refractory epilepsy patients implanted for the purposes of seizure localization. In contrast to ECoG, sEEG covers of cortical areas in sulci and subcortical areas posited to be involved in number processing by the TCM. The sEEG electrodes used currently, which are around 2 mm in diameter, measure brain activity much more locally than scalp electroencephalography (EEG); most of the activity is concentrated less than 1 cm away from the recording contact compared to large spread across cortical areas for EEG [17]. Stimulation can also be delivered using sEEG to perturb brain activity. Compared to TMS, sEEG has similar spatial specificity, both change activity most around 1 cm from the focus [18, 19]. There is a contrast between sEEG and TMS in the brain depth that is able to be stimulated, with TMS being limited to around 4 cm in depth [20] and sEEG is able to stimulate as deep as the contact is implanted. Comparing sEEG to fMRI, both record from the same sub-surface brain structures, however, fMRI measures blood oxygenation whereas sEEG is a direct measure of electrical neural communication. Practically, this is important because of reproducing brain-wide association studies using fMRI has been shown to take thousands of individuals [21] so having corroborating evidence from other neuroimaging modalities is essential, and sEEG can detect millisecond-time scale changes in neural activity whereas fMRI samples activity on the order of every two seconds. Thus, sEEG can detect the precise order of brain area activation as well as fast neural activity such as oscillations. In this study, we used sEEG to replicate the triple code involvement of the ventral temporal lobe, perisylvian region, and intraparietal sulcus and characterize the involvement structures outside these areas in numerical cognition. Drawing upon the TCM's postulates, we presented 13 subjects with a number recognition task of numerical stimuli that differed at the level of representation type (symbolic vs. non-symbolic) and mode of stimuli delivery (visual vs. auditory). Using a classification algorithm on sEEG recordings, we examined the extent to which number encoding brain regions aligned to the TCM. Additionally, this approach enabled us to explore neural substrates outside of this predicted framework.

## Materials and methods

### Participants

Thirteen patients with a diagnosis of medically refractory epilepsy were implanted with sEEG electrodes for the purpose of seizure onset zone localization. The sEEG electrodes used were 0.8 mm in diameter with a center-to-center pitch of 3.3 mm to 5mm between electrode contacts (PMT, Chanhassen, MN, USA). A total of 2,482 contacts were analyzed and distributed within each subject as shown in Fig 1. Pertinent demographic characteristics are tabulated in Table 1. Institutional Review Board approval was obtained at Oregon Health & Science University (STUDY00018870). All participants were over the age of 18 and provided written informed consent. Informed consent was obtained under the Declaration of the Principles of Helsinki.

### Behavioral task

Patients performed a passive numerical recognition task on a laptop (Fig 2). During the task, patients were presented with a number quantity from one to nine in one of four different representation formats. Two of the four were auditory (spoken number, sequential beeps) while the remaining two were visual (Arabic numeral, assortment of dots). These representation formats were designed to reflect common types of number stimuli encountered in daily life. At the start of each trial, a fixation cross was shown for between 500 and 1500 ms, chosen uniform randomly, to orient the patient's attention. The number stimulus was then presented for 1000 ms with an inter-trial period of 1500 to 3500 ms chosen uniform randomly. All 13 patients were presented with 200 number trials randomly shuffled so that there were 40 presentations of each number modality. To ensure attentiveness, patients were presented with catch trials on 10% of trials. They were prompted to press the left arrow key when the number was odd and the right arrow key when the number was even. The task would not proceed until patients responded to the catch trial. The code to administer the task is available at https://github.com/alexrockhill/numbers.

The task was administered using a custom jsPsych script implemented through a web browser [22]. The laptop was placed in front of patients on a table comfortably positioned over their lap. Trials were synchronized to intracranial electrophysiology using a photodiode connected into the same amplifier as the sEEG data and attached to either the right or left corners of the laptop screen [23]. Trials during corrupted photodiode events were excluded as accurate timing of intracranial electrophysiology changes could not be ascertained without photodiode synchronization. All participants included were right-hand dominant and responded to catch trials accordingly.

### Electrode localization

To determine sEEG electrode positions, preoperative stereotactic T1 and T2 magnetic resonance (MR) were registered to postoperative computerized tomography (CT) imaging studies with MNE-Python [24, 25]. Anatomic labels were assigned to contacts using the Desikan-Killiany atlas label of each patient's Freesurfer reconstruction [26]. Contact locations were warped to a template brain (cvs_avg35_inMNI152) to standardize contact positions across patients and allow between patient comparisons. Task-related cue and response events were synchronized using differences in time stamps recorded by the task computer relative to the time that the fixation stimulus was displayed which was synchronized by the photodiode. Using MNE-Python, time-frequency spectrograms for each event were computed using the Morlet wavelets method with frequencies from 1 to 250 Hz. After bandpass filtered between 0.1 and 40 Hz,

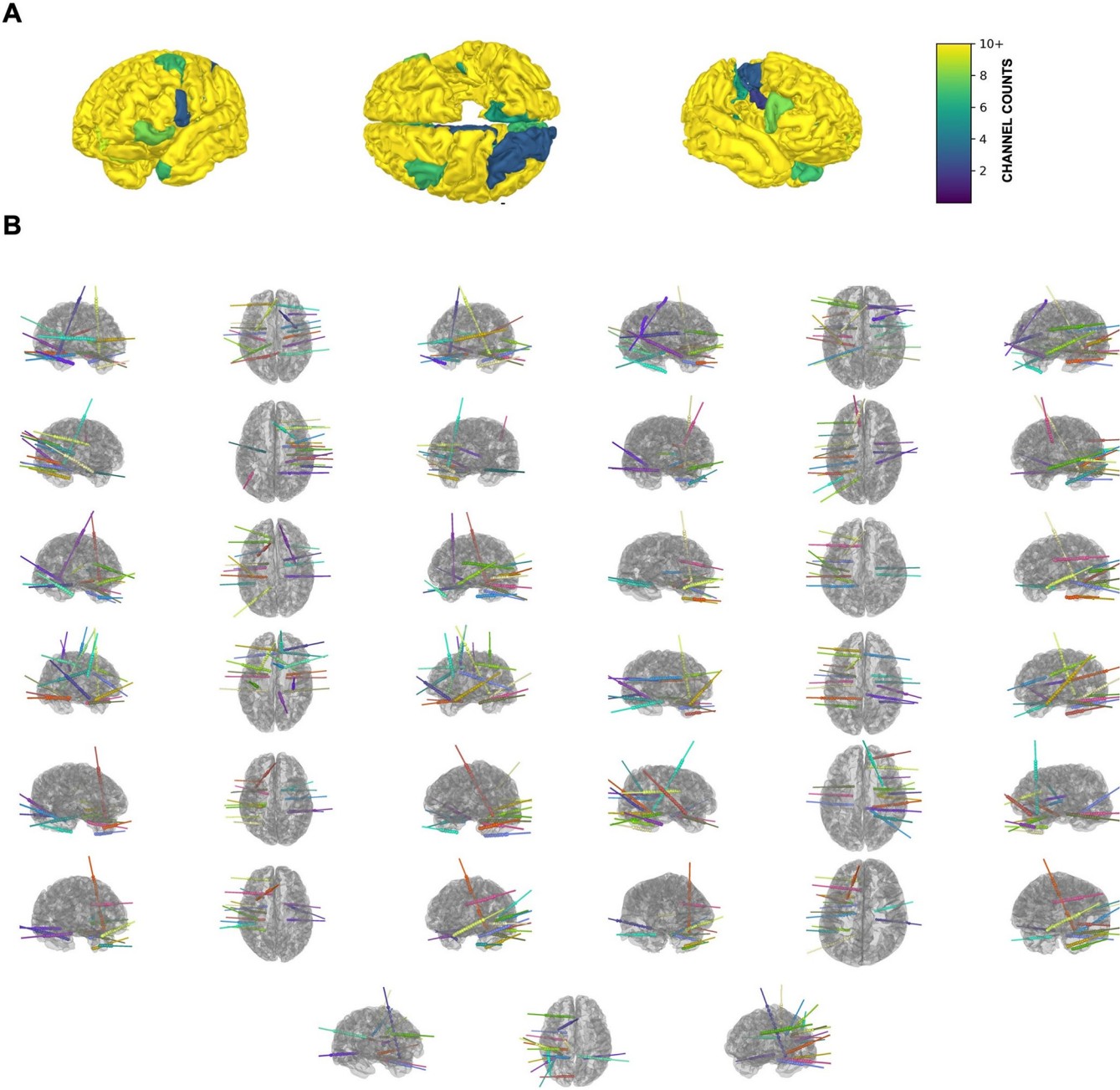

**Fig 1. Distribution and localization of sEEG electrodes.** Electrode localization (A) Heat map 94 of sEEG electrode contact distribution (B) sEEG implantation layouts for each of the 13 patients.

voltage time-series signals were appended to the bottom of each spectrogram to account for event-related potentials in our classification analysis.

## Pre-processing

Recordings from sEEG channels were average re-referenced, which has been shown to increase sensitivity to high-frequency broadband activity which is correlated with single unit activity

**Table 1. Subject demographics.**

| Subject | Age | Sex | Dominant Hand |
|---|---|---|---|
| S72 | 47 | female | right |
| S73 | 25 | female | right |
| S78 | 34 | female | right |
| S79 | 30 | male | right |
| S80 | 20 | male | right |
| S82 | 43 | male | right |
| S83 | 43 | female | right |
| S84 | 25 | male | right |
| S85 | 69 | male | right |
| S86 | 21 | male | right |
| S90 | 33 | male | right |
| S91 | 48 | male | right |
| S92 | 44 | female | right |

[27]. Trials where peak-to-peak amplitude at any recording contact was greater than a global rejection threshold estimated by autoreject [28].

## Classification

We leveraged a classification analysis similar to Rockhill et al. 2023 to determine the classification accuracy of sEEG channels, parcellated brain regions, and spectral features for all numerical stimuli (Fig 2). Number trial spectrograms consisted of the period -1 s prior to number stimulus presentation to 1 s afterward. These spectrograms were classified differently from inter-trial interval spectrograms of the same duration. Due to the size of the spectrogram data relative to the number of trials, the data had to be dimensionality reduced by principal component analysis (PCA) in order for the SVM to converge on classification that generalizes to unseen data [25]. Training spectrograms were dimensionally reduced with PCA. The first 50 principal components were used as inputs into a linear support vector machine (SVM) classifier deployed with scikit-learn [29]. Six-fold cross-validation was used. A binomial distribution with a probability of 0.5 and with the number of observations matching the number of presentations of each number modality was used as the null distribution. Coefficient matrices from the SVM were validated using a one sample cluster permutation test with a significance threshold set at 99% of a T-distribution (alpha = 0.01). Within sEEG channels, clusters were deemed statistically significant if their T-statistics were greater than 99% of permuted clusters. With 40 trials per number modality, we were powered at 79.97% to detect a large effect size of 0.4 between the stimulus and inter-trial interval conditions.

## Results

### SVM classifier accuracy

Our linear SVM successfully classified spectrograms during number stimuli and specific representation formats from those during the inter-trial interval. With an alpha threshold of 0.01 relative to the null distribution, we identified contacts with statistically significant classification probabilities. For our four representation formats, 360, 781, 962, and 342 contacts had significant classification values during Arabic, beeps, spoken, and dots, respectively, out of 2,567 total contacts (Fig 3). There were 74 contacts that were significant for all four representation types.

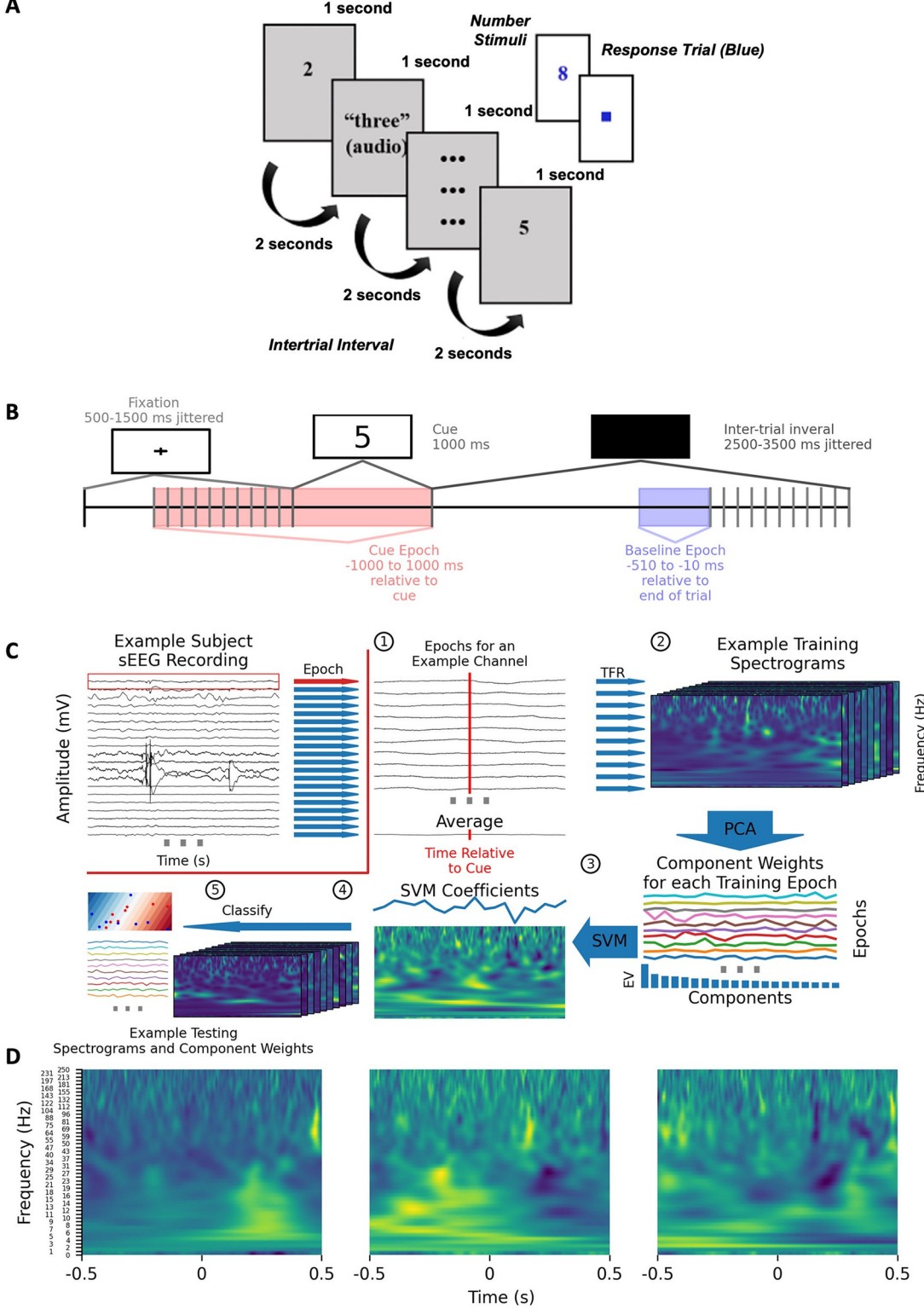

**Fig 2. Task paradigm and classification algorithm.** (A) The numerical task paradigm showing the timing of the task, examples of stimuli shown to patients and how catch trials were presented (blue). (B) The task timing used for the classification is shown. The jittered event timing avoids patients anticipating upcoming trials. The choice of a baseline period allowed at least 1990 ms after the last trial for brain activity to return to baseline. (C) a schematic of the classification analysis pipeline using an example contact is shown. This process is iteratively performed for all sEEG channels. (1) Continuous data is divided into epochs and (2)

decomposed into spectrograms. (3) PCA is fit to the spectrograms to reduce the dimension of the data. For each epoch, there is a weight for each PCA component, and the explained variance is shown as a bar plot beneath the line plot of weights for that component. (4) An SVM was fit to the PCA component weights, beneath, the projection back into spectrogram space is shown by multiplying the principal component weights by the SVM coefficients and summing the output. (5) Finally, the PCA component weight for the test trials were classified as being during the stimulus period of interest or inter-trial interval. The linear decision boundary for the first 2 principal components is shown as an example. (D) Three example principal components are shown projected back into time-frequency space showing "eigenspectrograms" which are spectrograms with important features for the classification task. This is done my multiplying each of the first three PCA components with the SVM coefficients to project them back into spectrogram space (like in (4) but not averaged).

### All number formats vs. Inter-trial period

We began by looking at spectrogram classifications that were significant across number representation formats compared to spectrograms during the inter-trial interval. The SVM classification demonstrated that left parietal regions and left lingual cortex had the highest classification accuracy (Fig 4A). The three contacts with the highest classification value across all number stimuli are presented on Fig 4B. The first was in the left lingual cortex. The second was in right fusiform gyrus. The third was in the superior frontal cortex. The contacts found to be significant for Arabic, dots, spoken, and beeps number representation formats were observed to be distributed in a fronto-parietal-temporal network that closely matches the TCM.

### Classification of auditory representation formats

SVM classification of beep spectrograms showed that bilateral superior temporal and right inferior parietal cortex had robust classification values (Fig 5A). Left putamen had classified with high accuracy as well (Fig 5A). Of the three contacts with the highest classification accuracy, two were in the right superior temporal gyrus while the third was in the left superior temporal gyrus (Fig 5B). Gamma and theta to low beta increases were used for classifying beeps in the two right superior temporal gyrus contacts. These changes were also present and used for classification in the left superior temporal gyrus contact, and both had similar timing with most activity occurring before the end of the audio presentation at 0 s.

Similar to the classification of beeps, spoken number trials demonstrated the best classification value in the bilateral superior temporal cortices (Fig 6A). Within the subcortex, the left putamen demonstrated a robust classification value similar to that of the superior temporal cortices (Fig 6A). The three contacts of highest accuracy were found in left superior temporal cortex(Fig 6B). High-frequency broadband increases were used to classify spoken number trials within superior temporal cortices. Additionally, sustained theta and alpha decreases were observed to be important for classification within both superior temporal cortex contacts.

### Classification of visual representation formats

The SVM classification of spectrograms during Arabic numeral trials indicated that the frontal lobes, left parietal lobe posterior to the postcentral gyrus and bilateral inferior temporal gyrus had robust classification value (Fig 7A). The three contacts of highest classification accuracy were localized in the temporal-occipital junction, middle temporal cortex, and the superior parietal lobule (Fig 7B). Decreases and increases in alpha and increases in high-frequency broadband activity were most important for classifying Arabic numerals for the contacts located in these areas.

Despite having fewer contacts than other regions, the left parietal cortex had the best classification value when the SVM classified dot number stimuli spectrograms compared to inter-trial interval spectrograms (Fig 8A). The bilateral frontal lobe had robust classification value as

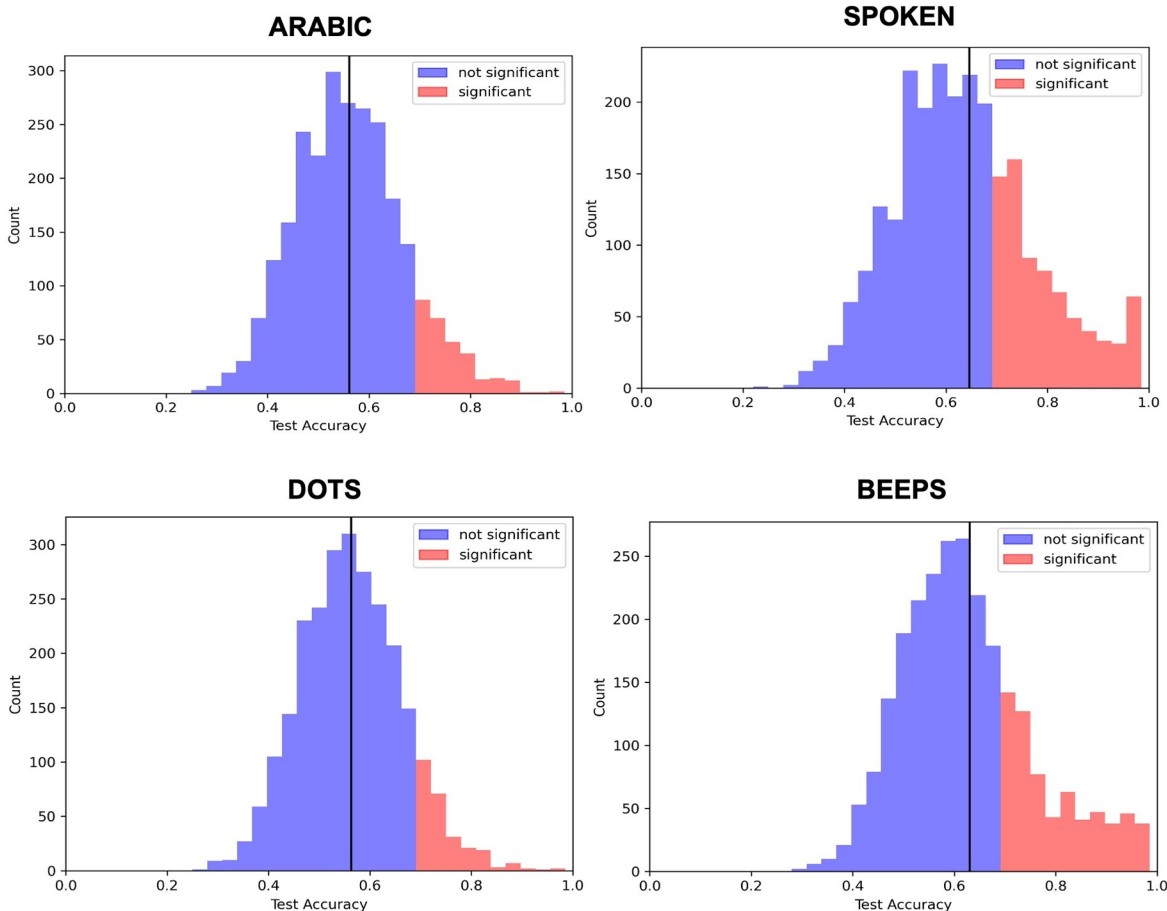

**Fig 3. Distribution of sEEG channel classification accuracy across different numerical representations.** Histograms presenting the counts of sEEG channels with significant (red) and non-significant (blue) classification accuracy at alpha = 0.01 relative to a null binomial distribution across all contacts are shown. One sample t-tests comparing the difference between the test distribution and chance resulted in p<0.001 for all comparisons. The distribution of significant and non-significant sEEG channels are presented for each number representation format.

well. The three contacts with the highest classification accuracies were in and around bilateral superior parietal lobule (Fig 8B). Alpha and beta decreases and high-frequency broadband increases were important for classifying dot trials.

## Spectral features with high classification value for number stimuli

To better understand the SVM classification coefficients, we generated feature maps showcasing the relative abundance of statistically significant time-frequency clusters and the proportion of positive significant clusters to determine the directional patterns of time-frequency cluster changes (Fig 9). Then, we assessed the classification value of these cluster changes. Across all number representation formats, significant clusters within the theta to beta frequency ranges were well represented. To a less robust degree, spoken number and beeps demonstrated a number of significant high-frequency broadband clusters as well. The theta and beta clusters generally increased in power for all representation formats with the exception of spoken number, where beta clusters demonstrated an initial increase before a decrease. For beeps and spoken number formats, there were significant high-frequency broadband clusters with increased power during the experimental condition compared to the inter-trial intervals.

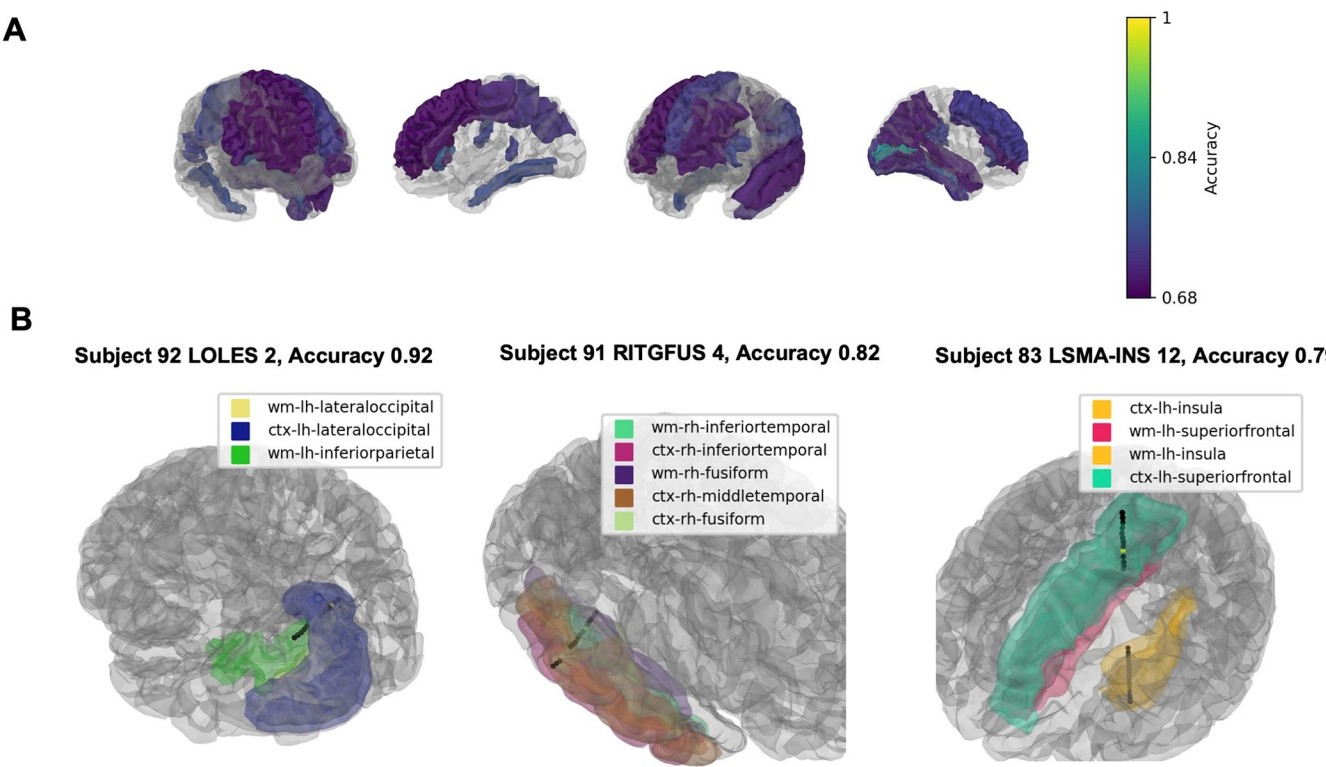

**Fig 4. Linear SVM classification of spectrograms from all number trials versus inter-trial interval spectrograms.** Linear SVM classifications that were significant for all number representation formats versus the inter-trial interval. Classification values of parcellated brain regions are presented with a gradient (Yellow; 1 = 100% classification accuracy, Purple: 0.68 = 68% classification accuracy) (A). The three contacts with the highest classification value are shown in (B).

Increases in high-frequency broadband power had the highest classification accuracy. Alpha and beta were less closely associated with high-accuracy classifications but were still important to classification which were unlikely to happen by chance.

## Discussion

### Brain regions encoding number processing

Using a linear SVM classifier with PCA, we identified cortical and subcortical structures, individual contacts, and spectral patterns with high classification value during our number stimuli trials. Overall, our findings align with results from prior fMRI and ECoG investigations into number cognition, [6, 8, 11, 31] and by extension, are congruent with existing neuroanatomical models of number cognition.

When number stimuli, irrespective of representation format, were classified against the inter-trial period, we found evidence of bilateral parietal lobe engagement and robust engagement of the left inferior parietal cortex, which aligns with the postulates of the TCM (5). Intraparietal sulcus is hypothesized to be responsible for encoding an innate, conserved sense of non-abstract number processing [5, 6]. Previous fMRI studies have implicated the involvement of this region across a variety of number stimuli which has been replicated in ECoG studies [9, 30, 31]. We also observed that auditory numerical stimuli reliably engaged these putative cortical substrates. For both beeps and spoken number trials, the bilateral superior temporal cortices had high-accuracy classifications, suggesting preferential activation at these

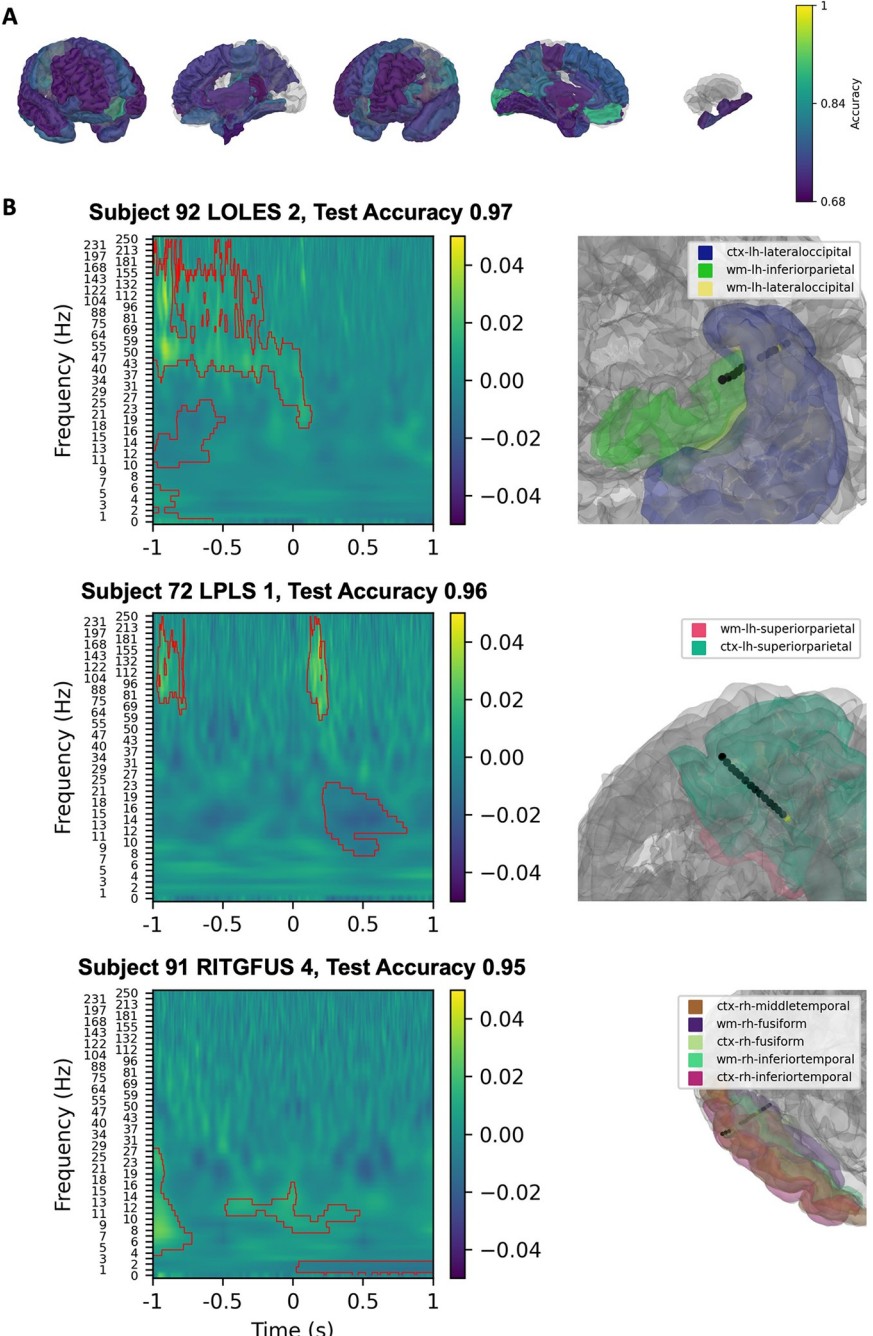

**Fig 5. Linear SVM classification of spectrograms from sequential beep trials versus inter-trial interval spectrograms.** Linear SVM classification of spectrograms from sequential beep trials versus inter-trial interval spectrograms. Classification values of parcellated cortical and subcortical brain regions are presented with a gradient (Yellow; 1 = 100% classification accuracy, Purple: 0.68 = 68% classification accuracy) (A). The three contacts with the highest classification value are shown in (B). Left panels show SVM coefficients from the spectrogram classification projected back to spectrogram space using the principal components with significant clusters shown by the red contours. Right panels show the location of the contact within parcellated brain regions.

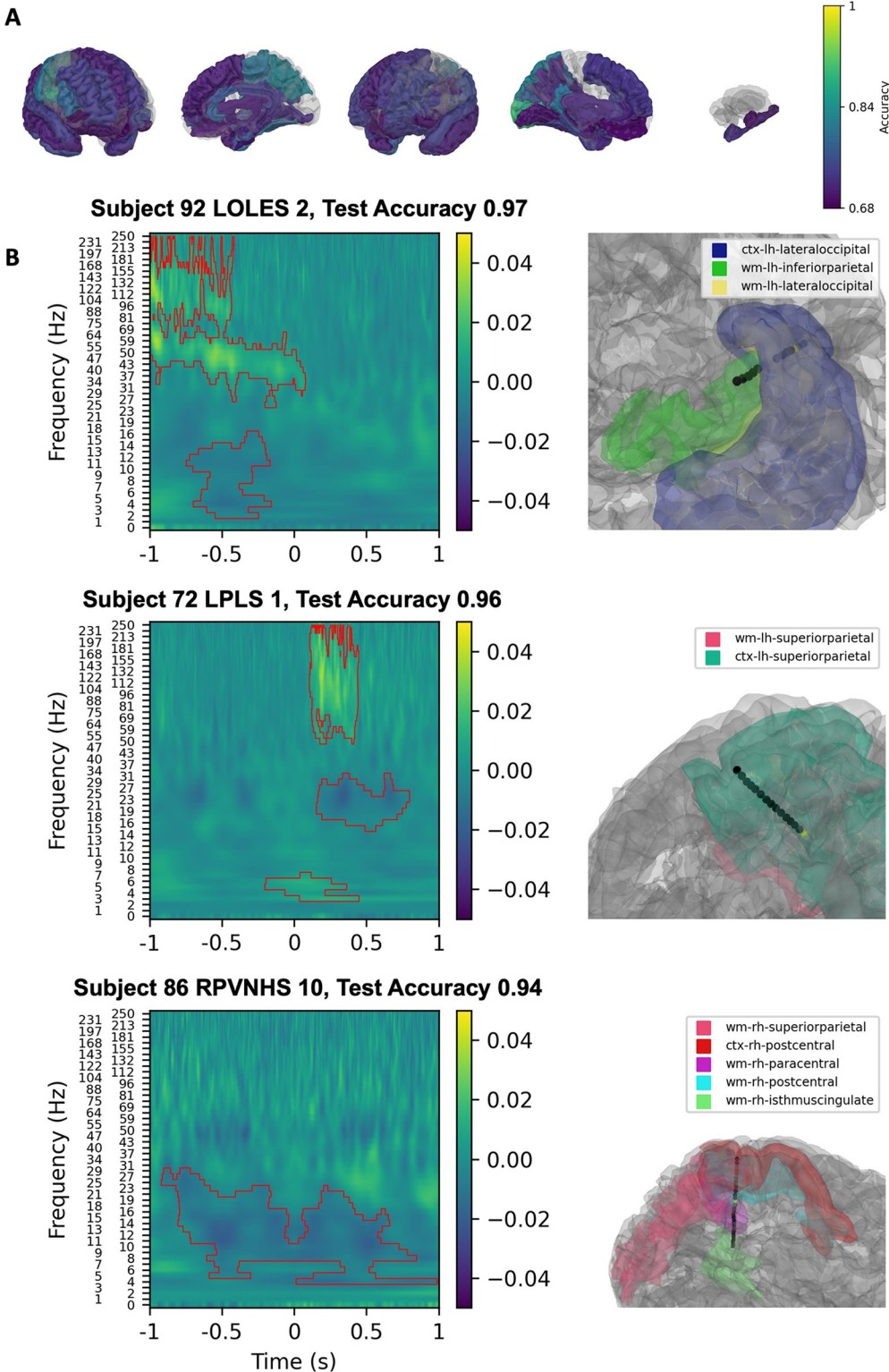

**Fig 6. Linear SVM classification of spectrograms from all spoken number trials versus inter-trial interval spectrograms.** Linear SVM classification of spectrograms from all spoken number trials versus inter-trial interval spectrograms. Classification values of parcellated cortical and subcortical brain regions are presented with a gradient (Yellow; 1 = 100% classification accuracy, Purple: 0.68 = 68% classification accuracy) (A). The three contacts with the highest classification value are shown in (B). Left panels show SVM coefficients from the spectrogram classification

projected back to spectrogram space using the principal components with significant clusters shown by the red contours. Right panels show the location of the contact within parcellated brain regions.

sites. This confirms known localizations of the eloquent human auditory cortex which have been implicated by fMRI analyses to underline auditory number processing [32–34].

Classification of the visual representation formats implicated similar regions to expected cortical regions of engagement. Prior ECoG analyses have identified a visual number form area (VNFA) within the fusiform and inferior temporal gyri that is preferentially engaged in response to Arabic numerals (10,11). During Arabic trials, we found evidence of this activation as well as frontal lobe involvement with evidence of inferior parietal and middle temporal cortical engagement. Our sEEG channels did not sample the VNFA densely, potentially explaining why contacts with high-accuracy classifications were not even more predominately located in this area. Finally, classification of dot trials indicated preferential engagement of the left parietal cortex, as expected, as well as the left frontal lobe. This aligns with the frontoparietal network of number cognition posited by Dehaene et al. and the hypothesized role of the parietal cortex put forth by the TCM [4, 5] with the notable difference that the right parietal sEEG channels had lower classification accuracy, questioning the whether the right-hemisphere dominance found in previous studies should be re-examined.

Interestingly, for auditory number representation formats, the left putamen had the best classification value of all subcortical structures. The classification accuracy of the putamen was comparable to that of putative cortical substrates of number processing. The superior classification of the left putamen compared to the right is unclear. However, this may be partially related to the hemispheric dominance of our patients as all were left hemisphere dominant. Although commonly associated with motor function as a component of the basal ganglia, the putamen is known to take part in higher-level cognitive functions as well [35–37]. Enhanced putaminal engagement during numerical processing has been demonstrated during magnitude evaluation and arithmetic tasks with functional imaging [12, 14]. Recently, using a high field MRI, investigators recently detected tuned neural responses to numerical quantities within the putamen during a tactile numerosity task [14], illustrating the pertinent role of the putamen in integrating and comprehending numerosity inputs.

Considering that neuroimaging studies have also demonstrated the role of the putamen in language cognition [35, 36], the engagement of the putamen during our number recognition paradigm touches on the potential interplay between number cognition and language. We did not attempt to compare neuroanatomic and electrophysiologic features of number stimuli to language ones in our study due to time constraints. There is spirited debate, particularly within neuropsychology and social sciences literature, over whether the development of and capacity for numerical cognition is independent of language acquisition [38]. More recent evidence suggests that the development of large and exact numerosity representations is contingent upon access to language [39]. As a corollary, conceptualizing smaller and less precise quantities may be agnostic of language, which suggests that there are indeed numerosity specific, or language independent, neural substrates circuitry. Previous studies have demonstrated the involvement of the putamen in visual processing of Roman numerals [40], magnitude estimates of negative numbers [41] and haptic numerosity [11]. Our findings not only replicate the involvement of the putamen in processing of numerals but also provide detail about how the putamen responds to different modality presentations of numbers and the time-frequency characteristics of those responses. Disentangling numerical circuits from language ones could be of clinical value, especially in the setting of Gerstmann syndrome which is classically typified by acalculia. To this end, future studies should leverage structural and functional

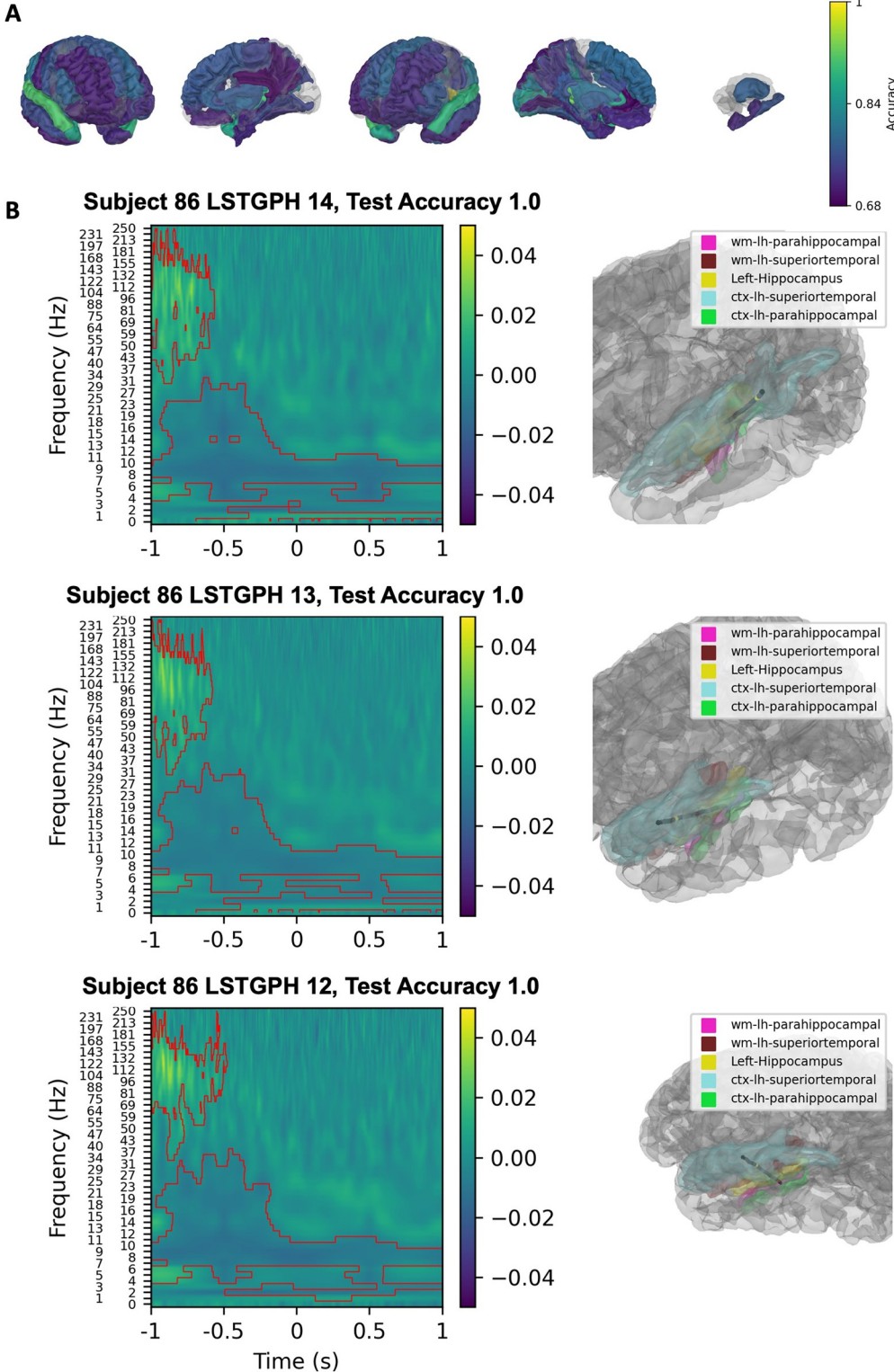

**Fig 7. Linear SVM classification of spectrograms from all arabic numeral trials versus inter-trial interval spectrograms.** Classification values of parcellated cortical and subcortical brain regions are presented with a gradient (Yellow; 1 = 100% classification accuracy, Purple: 0.68 = 68% classification accuracy) (A). The three contacts with the highest classification value are shown in (B). Left panels show SVM coefficients from the spectrogram classification projected back to spectrogram space using the principal components with significant clusters shown by the red contours. Right panels show the location of the contact within parcellated brain regions.

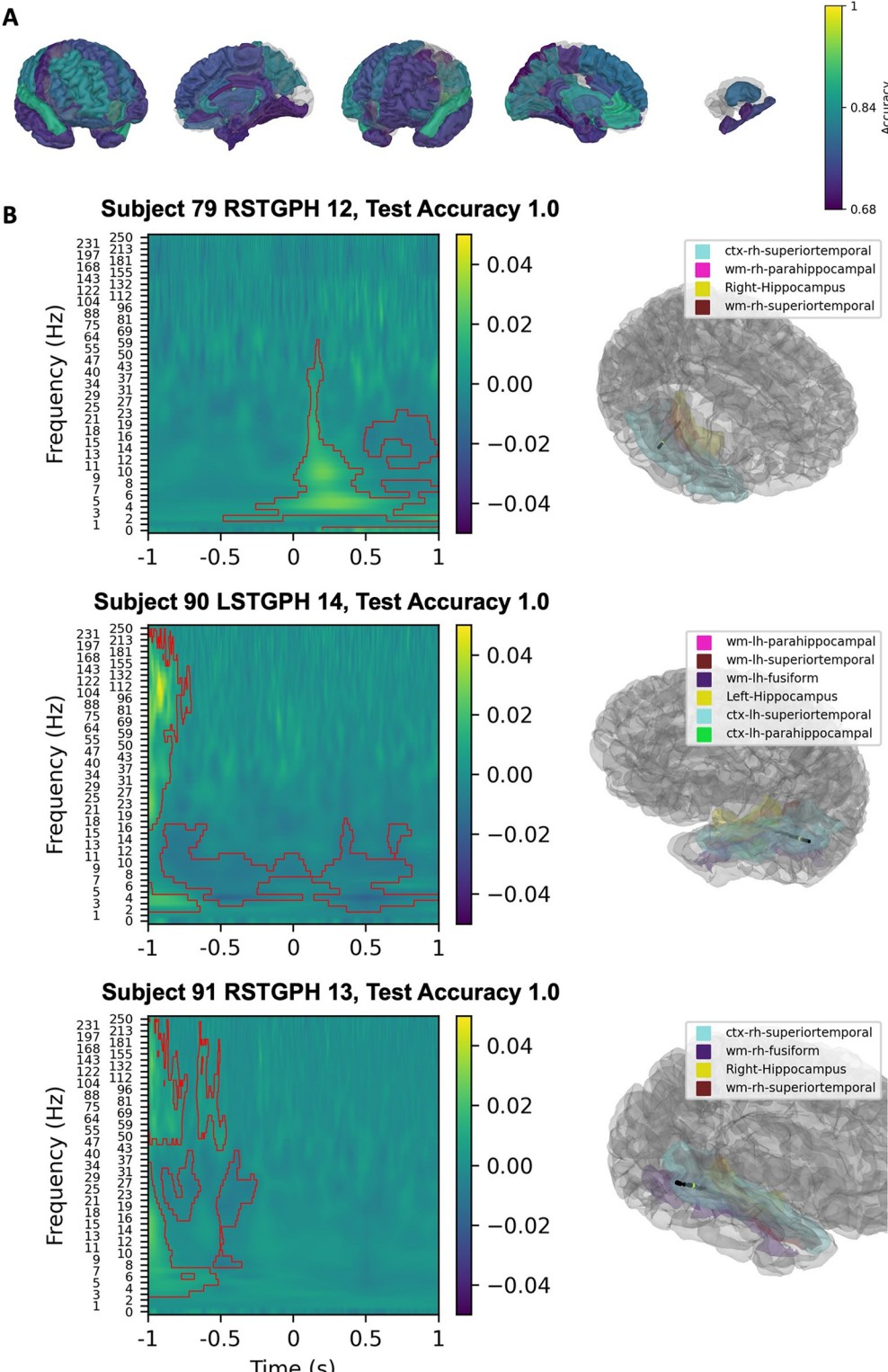

**Fig 8. Linear SVM classification of spectrograms from all assorted dots trials versus inter-trial interval spectrograms.** Classification values of parcellated cortical and subcortical brain regions are presented with a gradient (Yellow; 1 = 100% classification accuracy, Purple: 0.68 = 68% classification accuracy) (A). The three contacts with the highest classification value are shown in (B). Left panels show SVM coefficients from the spectrogram classification projected back to spectrogram space using the principal components with significant clusters shown by the red contours. Right panels show the location of the contact within parcellated brain regions.

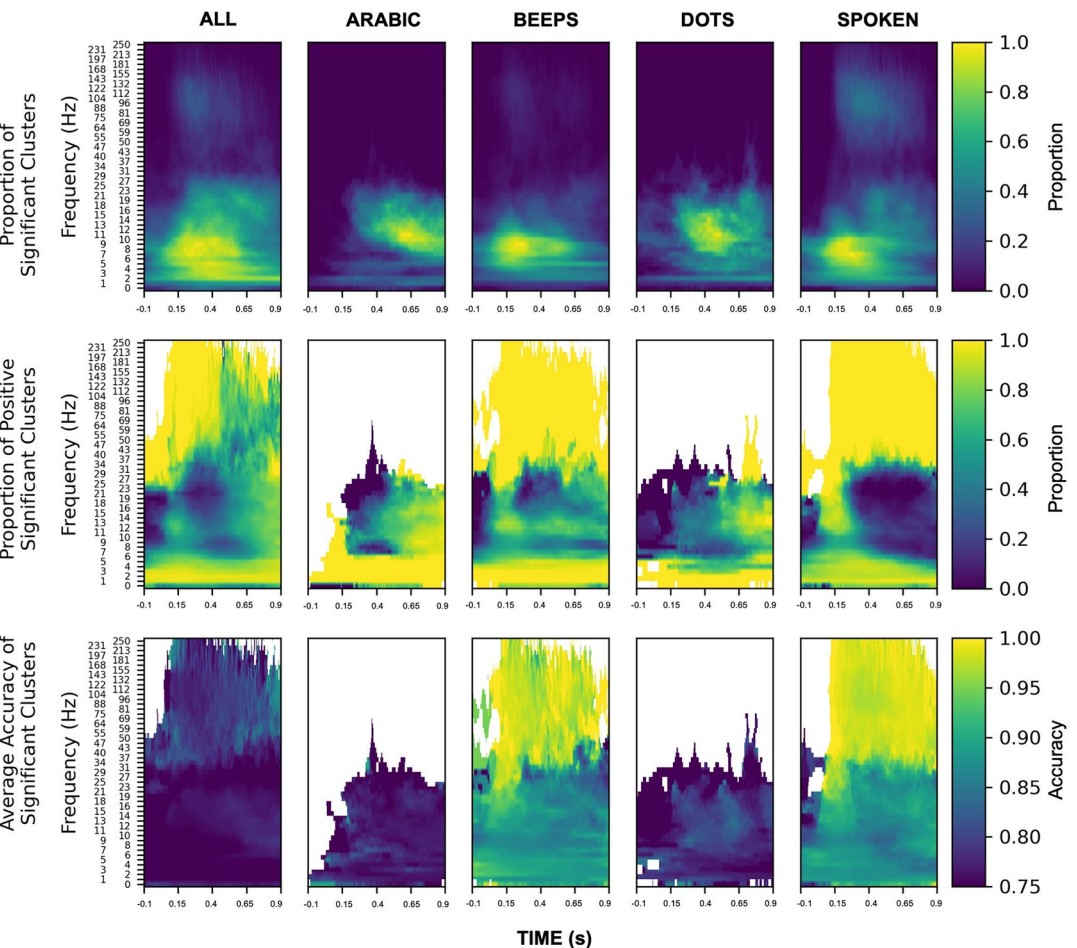

**Fig 9. Summary map of all contacts with statistically significant SVM classifications.** The top row presents the proportion of significant SVM clusters. It illustrates what proportions of channels are significant at that time-frequency ranging from 0 to 1. Yellow indicates a greater proportion of clusters while blue represents few clusters at specific time-frequency points. The middle row shows the directionality of spectral change for these significant SVM clusters during number trial conditions versus inter-trial intervals. It illustrates which proportions are positive ranging from 0 to 1. Yellow indicates an increase in time-frequency clusters during number trials versus inter-trial intervals while blue indicates a decrease in time-frequency clusters. White regions in the plot indicate that no directional change occurred. The bottom row presents the classification value of the directionality of SVM cluster changes. Yellow reflects high classification accuracy, while dark blue represents less robust classification accuracy. Time-frequency points with no significant classification value are white.

connectivity methods while directing specific attention towards the involvement of subcortical structures. The development of smaller scale and higher resolution recordings will be instrumental to these future investigations.

Because fMRI analyses hinge upon blood-oxygen-level-dependent (BOLD) signal changes, which are strongly coupled with high-frequency broadband activity and local field potentials [42], prior iEEG studies into number cognition have largely directed their attention to this band as a marker for synchronized neuronal activity [8–11]. While high-frequency broadband activity had strong classification value within our analysis, our time-frequency cluster feature maps suggest that power changes in lower frequency bands may also be of value in distinguishing numerical stimuli from non-numerical stimuli and characterizing specific representation formats. Future studies should attempt to further elucidate the role of theta, alpha, and beta frequency changes during number processing.

## Limitations

There were several limitations to our study. First, by only using sEEG, the spatial resolution of our cortical recordings was constrained because the placement of electrodes is solely determined by clinical purposes. Nonetheless, we were still able to associate number stimuli with structures that align with existing models of number cognition, specifically the TCM. Second, we chose to use a linear SVM with PCA as our classification method knowing that this may come at the sacrifice of classification accuracy. We opted against a more complex classification method with the intent of prioritizing interpretability over classification accuracy. Despite these limitations, to our knowledge, this is one of the first studies to utilize sEEG depth electrodes for the purpose of investigating human number cognition through sampling both cortical and subcortical structures.

## Conclusion

In conclusion, we used a machine learning classifier to identify cortical and subcortical substrates of human number cognition. Our findings support postulates of the TCM in number processing. However, we also determined that subcortical structures, particularly the putamen exhibited robust classification accuracy in response to numerical stimuli, thus expanding this framework. Analyses of spectral feature maps revealed that theta, alpha and beta frequency bands held greater than chance classification value and could be potentially used to characterize format specific number representations. We provide both neuroanatomical and electrophysiologic targets of interest that can be leveraged in future number cognition investigations.

## Author Contributions

**Conceptualization:** Ahmed M. Raslan.

**Data curation:** Alexander P. Rockhill, Hao Tan, Christian G. Lopez Ramos, Caleb Nerison, Maryam N. Shahin, Adeline Fecker.

**Investigation:** Caleb Nerison, Beck Shafie, Maryam N. Shahin, Adeline Fecker, Mostafa Ismail, Ahmed M. Raslan.

**Methodology:** Alexander P. Rockhill.

**Project administration:** Alexander P. Rockhill, Hao Tan, Christian G. Lopez Ramos, Caleb Nerison, Ahmed M. Raslan.

**Resources:** Kelly L. Collins, Ahmed M. Raslan.

**Software:** Alexander P. Rockhill.

**Supervision:** Kelly L. Collins, Ahmed M. Raslan.

**Writing – original draft:** Hao Tan, Christian G. Lopez Ramos.

**Writing – review & editing:** Alexander P. Rockhill, Hao Tan, Christian G. Lopez Ramos, Beck Shafie, Maryam N. Shahin, Adeline Fecker, Mostafa Ismail, Daniel R. Cleary, Kelly L. Collins, Ahmed M. Raslan.

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
