## [Decision Letter · Decision Letter 0]

1 Jul 2024

PONE-D-24-19042Investigating the Triple Code Model in Numerical Cognition Using Stereotactic ElectroencephalographyPLOS ONE

Dear Dr. Lopez Ramos,

Thank you for submitting your manuscript to PLOS ONE. After careful consideration, we feel that it has merit but does not fully meet PLOS ONE’s publication criteria as it currently stands. Therefore, we invite you to submit a revised version of the manuscript that addresses the points raised during the review process.

We look forward to receiving your revised manuscript.

Kind regards,

Kiyoshi Nakahara, PhD

Academic Editor

PLOS ONE

Journal Requirements:

Reviewers' comments:

Reviewer's Responses to Questions

**Comments to the Author**

1. Is the manuscript technically sound, and do the data support the conclusions?

Reviewer #1: Partly

Reviewer #2: Yes

2. Has the statistical analysis been performed appropriately and rigorously? 

Reviewer #1: No

Reviewer #2: Yes

3. Have the authors made all data underlying the findings in their manuscript fully available?

Reviewer #1: Yes

Reviewer #2: Yes

4. Is the manuscript presented in an intelligible fashion and written in standard English?

Reviewer #1: Yes

Reviewer #2: Yes

5. Review Comments to the Author

Reviewer #1: This study uses stereotactic electroencephalography (sEEG) to explore the neural mechanisms of numerical cognition in humans. By applying the Triple Code Model, which posits that different brain regions process visual Arabic, auditory verbal, and nonsymbolic analogue numbers, the authors identified a fourth region in the subcortex, the putamen. The brain cortex was extensively covered by numerous sEEG electrodes, and this investigation is potentially important for advancing our understanding of numerical cognition. However, several concerns need to be addressed to clarify the impact of this study.

Major comments:

1. The resolutions of the figures are too low to see details or read labels and legends, even in the downloaded tiff files. Therefore, I cannot verify if the authors' descriptions in the manuscript are accurately represented by the figures.

2. As discussed in the manuscript, the involvement of the putamen has been suggested in several prior fMRI studies (e.g., Masataka et al., 2007, Brain and Language; Gullick et al., 2012, NeuroImage; Hofstetter & Dumoulin, 2021, NeuroImage). Early studies (e.g., Dehaene and Cohen, 1997, Cortex) also suggested the involvement of the basal ganglia, including the putamen. Given this background, the novelty of this study is blurred, and the introduction should better justify the need for sEEG over fMRI to understand subcortical contributions. To emphasize the advantages of sEEG, discuss the contributed latencies and frequencies in more detail. Conversely, the current discussion, which mentions the relations between fMRI BOLD and high-gamma, weakens their findings because their new findings of high-gamma contribution can be well predicted in the literature.

3. Although the authors claim that their cortical findings are consistent with the TCM, it is difficult to understand this consistency. The authors list multiple brain regions in their results but do not adequately discuss how these regions align with or differ from the TCM and prior studies.

4. The authors used machine learning classification within stimulus presentation forms, contrasting with inter-trial intervals. This approach may not be optimal for identifying critical brain regions for different numerical representation forms. Instead, classification across all stimulus forms would better highlight the differences in neural processing. Additionally, the current method may not effectively separate perceptual processing from numerical cognition. Including catch trials with fake digits, words, and nonsymbols without numerical meaning could address this issue.

5. The procedure of the SVM classification with PCA has not been well-described, leading to confusion about the outputs of the analyses. The authors first computed principal components of the spectrograms and trained a component-based coefficient matrix consisting of time, frequency, and components. They then computed a channel-based coefficient matrix by applying the inverse matrix of the PCA. Finally, they applied the channel-based coefficient matrix to the test data and obtained a classification accuracy for each stimulus form. However, it seems classification accuracies were computed for each sEEG channel, as shown in Figure 3. This raises the question of how these data were obtained, given that channels were decomposed into principal components, making classification within channels seemingly impossible. Furthermore, the “null distribution” depicted in Figure 3 is not explained in the manuscript. The authors mention using a “one-sample cluster permutation test” on the coefficient matrix of the SVM classification, but there is no explanation of this process either. If the “null distribution” was computed in the permutation test, it should be across clusters rather than channels. This needs to be clarified.

6. Were the classification accuracies cross-validated? The authors mention a "70/30% training-test split." If this means a single random split, I recommend applying cross-validation by repeating the computation multiple times to ensure robustness.

Minor comment:

1. The authors interpret their finding of the left parietal cortex's role in processing numerical dots as aligning with "the hypothesized role of the left parietal rgcortex put forth by the TCM." Did Dehaene specifically limit this role to the left side? Also, "rgcortex" appears to be a typo.

2. Figure 9 lacks an explanation of the color-code used.

3. Line 52: "Triple Code Model" (TCM) seems to be incorrectly abbreviated as "TMC."

4. There are inconsistencies in spelling "sEEG contacts" and "sEEG channels."

Reviewer #2: This is a very interesting study looking into the triple-code model, providing a neural basis for the model, and understanding it further. The rationale and the purpose of study is clear and valid. I think the parts explaining the electrodes are strong and results are nicely explained for each format. Additionally, discussing the limitations faced during the study were helpful and act as a positive thing, as it could provide valuable insights for future research in this area.

The literature review could help tell more about the sEEG and why it might be better than other mediums such as TMS and ECoG. I think knowing a bit more about sEEG in the introduction will help the reader. It is not mentioned but I understand it is a single session study, with its duration, time, and number of trials for each task not known. Providing more information about the methodology, including the study design and parameters, would enhance the clarity and replicability of the research. The results lack statistical analysis of the data. It is more demonstrated using pictures but would be more significant with statistical analysis. I would recommend showing some power statistics for the results, which could provide more credibility to the findings and ensure the study's conclusions are supported by robust data analysis.

Overall, this study contributes to our understanding of the triple code model and its neural basis. It might open new avenues for further research in this field, potentially leading to breakthroughs in understanding numerical cognition.

6. PLOS authors have the option to publish the peer review history of their article (what does this mean?). If published, this will include your full peer review and any attached files.

Reviewer #1: No

Reviewer #2: No

---

## [Author Response · Author response to Decision Letter 0]

17 Aug 2024

Thank you to both of the reviewers for your time. Both of your comments were very insightful and helped improve the quality of the manuscript. We have done our best to incorporate the changes you suggested and to provide more detail on the points that you commented on, especially in the few areas where it was not possible or practical to revise the manuscript to completely satisfy the critique. We hope that, based on the point-by-point responses below and the highlighted changes to the manuscript, we have revised our manuscript to meet the high standards of PLOS ONE in your expert opinions.

Response to reviewer 1: 

1. Low figure Resolutions

Response: We have replaced all figures with high resolution versions. 

2. 

a. Several fMRI studies (e.g., Masataka et al., 2007, Brain and Language; Gullick et al., 2012, NeuroImage; Hofstetter & Dumoulin, 2021, NeuroImage) have suggested the involvement of putamen which blurs the novelty of this study.

Response: We agree with the important precedence of these studies and have added citations for them in the manuscript. We would defend the novelty of these claims based on the time-frequency resolution of electrophysiological measurements and the important differences in task design, including our inclusion of a wide array of modalities in which the numbers were presented. The addition to the manuscript in the conclusion is included below (line 385-390):

Previous studies have demonstrated the involvement of the putamen in visual processing of Roman numerals (Masataka, 2007), magnitude estimates of negative numbers (Gullick, 2012) and haptic numerosity (Hofstetter & Dumoulin, 2022). Our findings not only replicate the involvement of the putamen in processing of numerals but also provide detail about how the putamen responds to different modality presentations of numbers and the time-frequency characteristics of those responses.

b. [The] introduction should better justify the need for sEEG over fMRI to understand subcortical contributions. To emphasize the advantages of sEEG, discuss the contributed latencies and frequencies in more detail

Response: We added the following sentences in the introduction to address this point. Given recent and past findings questioning reproducibility in the field of fMRI neuroimaging, we think that the importance confirmation by another neuroimaging modality is justified. And sEEG detects fast changes in neural activity like oscillations, providing additional information to fMRI. Thank you for this comment, adding these changes makes it more clear to the reader the importance of this research. (line 83-91)

Comparing sEEG to functional magnetic resonance imaging (fMRI), both record from the same sub-surface brain structures, however, fMRI measures blood oxygenation whereas sEEG is a direct measure of electrical neural communication. Practically, this is important because of reproducing brain-wide association studies using fMRI has been shown to take thousands of individuals (Marek, 2022) so having corroborating evidence from other neuroimaging modalities is essential, and sEEG can detect millisecond-time scale changes in neural activity whereas fMRI samples activity on the order of every two seconds. Thus, sEEG can detect the precise order of brain area activation as well as fast neural activity such as oscillations. 

c. Discussion mentions the relations between fMRI BOLD and high-gamma, weakens the findings because new findings of high-gamma contribution can be well predicted in the literature

Response: This is a very reasonable critique. While fMRI is incredibly valuable and has made the most progress to date in understanding the neural correlates of numerical cognition, fMRI is not perfect and sEEG nicely complements those imperfections with better time-resolution, including differentiating when the high-gamma activity occurred relative to the presentation of the stimulus, and high signal-to-noise allowing us to present many different stimulus modalities in a relatively short period of time. Thus, we posit that, although the information presented in this study is not as foundational as early fMRI studies, the novel details from this study are still important and may be essential pieces of information for piecing together the more nuanced aspects of numerical cognition.

3. Difficult to understand the consistency with TCM without discussing how these regions align with or differ from TCM and prior studies 

Response: The aim of this study was to re-examine the basis of the triple code model as originally postulated using sEEG. Thus our focus was on the very basic replication of inferior temporal gyrus as visual number form area, superior temporal gyrus as verbal number language processing and intraparietal sulcus as non-symbolic number processing area and to explore how structures outside these triple code areas are involved. Because we did not specifically set out to study the involvement of the putamen or other subcortical structures, rather to see which structures were found to be involved based on the significance of statistical quantification of the classifications, we decided not to revise the introduction to include these structures so as to attempt to present our scientific process as it actually occurred. We did clarify this point in the end of the introduction so that this lack of clarity will not cause issues for readers. We also added comparisons to the original triple code model as well as later work extending this model in the discussion.

This aligns with the frontoparietal network of number cognition posited by Dehaene et al. and the hypothesized role of the parietal cortex put forth by the TCM (Dehaene & Cohen, 1995; Dehaene, 2003) with the notable difference that the right parietal sEEG channels had lower classification accuracy, questioning the whether the right-hemisphere dominance found in previous studies should be re-examined. 

4. Current method may not be optimal for identifying critical brain regions from different numerical representation forms, and classification across all stimulus forms would better highlight the differences in neural processing

Response: This is an excellent point, but unfortunately, the resolution of brain activity for current sEEG recording technology is apparently not up to this level of specificity. We analyzed but did not present decoding the numerical quantity and <= 3 vs. > 3 based on previous work by Dehaene showing behavioral differences in the recognition of three or fewer compared to more than three but no channels decoded these features with significant accuracy. This is likely due to the relatively small number of neurons that are tuned to specific numerical quantity which are not numerous or synchronous enough to be detected by sEEG, compared to the greater number of neurons that are tuned to number, possibly with an ideal response to a particular number, such that for numeric stimuli there is greater synchronous activation compared to during the inter-trial interval. Our group is involved in developing higher-resolution recordings for this purpose and so we have added this future direction to the discussion as well as adding transparency of analyses with null results.

5. Comment 5:

a. A. SVM classification with PCA has not well described leading to confusion about the output of analyses

Response: The SVM classification using PCA components is quite complicated, and we revised the caption for Figure 2 to ameliorate this issue. In brief, this technique was used because the simplicity of the SVM makes it interpretable but it is not able to converge due to the size of the spectrogram data relative to the number of trials. Dimensionality reduction in the form of PCA, allows the SVM to converge efficiently and yields stable, generalizable results. Thank you for bringing this lack of a high-level overview of this method to our attention, we have also a similar summary to the methods of the text for the reader.

b. The explanation of null distribution 

Response: The null distribution is a binomial with probability 0.5 corresponding to the 0.5 probability the stimulus-inter-trial interval classification being correct due to chance given the number of trials that the particular number modality was shown. This explanation was added in the caption of Fig 3 and in the Methods section. This distribution was found to be nearly identical to dividing up the inter-trial interval into two non-overlapping segments and classifying whether the data came from the first or second segment. In an earlier version of the results, the null classification distribution was used, but in the figures in the current version of the manuscript, the binomial distribution was used. This has been corrected in the Methods section of the text.

6. Clarify whether classification accuracies were cross validated. Recommendation: applying cross validation by repeating the computation multiple times to ensure robustness

Response: This is an excellent point to ensure robust statistical analysis. In this case, the results of the cross-validation analysis were very similar as shown below. Consequently, we added a sentence in the methods endorsing cross-validation and explaining that the results were similar in this case.

– please refer to "Response to Reviewers" document to see corresponding figures. 

A 70/30% training-test split was used. This yielded results that were found to be similar to using six-fold cross-validation.

Minor comments:

1. Comment 1:

a. Interpretation of left parietal cortex in processing numerical dots as aligning with “the hypothesized role of the left parietal [cortex] put forth by the TCM." Did Dehaene specifically limit this role to the left side?

Response: Dehaene suggested that both hemispheres are involved and, in fact, that the right side may even be dominant (excerpt below). Since we found greater involvement in the left hemisphere, we added an explanation of this important difference in the text. Thank you for catching this!

b. "rgcortex" appears to be a typo.

Response: “rgcortex” was corrected to “cortex”.

2. Figure 9 lacks explanation of the color code used

Response: The cluster permutation test between the inter-trial interval and stimulus period is difficult to interpret because it has both a sign (increased or decreased during the stimulus) and a proportion of channels for which that particular time-frequency is significant. To show both pieces of information, the first row what proportion of channels are significant at that time-frequency ranging from 0 to 1, and the second row shows which proportion are positive ranging from 0 to 1. This way, if there are time-frequencies that increase in some locations and decrease in others, they will still be shown in yellow in the first row, they will just be an intermediate green color in the second row. This explanation has been added to the caption and colorbars have been added to the figure.

3. Line 52: "Triple Code Model" (TCM) seems to be incorrectly abbreviated as "TMC."

Response: “TMC” was corrected to “TCM”.

4. There are inconsistencies in spelling "sEEG contacts" and "sEEG channels"

Response: “sEEG contacts” were changed to “sEEG channels”. 

Reviewer 2:

1. The literature review could help tell more about the sEEG and why it might be better than other mediums such as TMS and ECoG.

Response: We added details to the introduction about the specific brain areas that sEEG records from that ECoG does not, and we added details about the specificity of recording and stimulation of sEEG relative to scalp EEG and TMS in the third paragraph of the Introduction.

2. Providing more information about the methodology, including the study design and parameters, would enhance the clarity and replicability of the research

Response: We added details about the task such as the exact proportion of catch trials and the duration of the stimulus period. Furthermore, we added a link to the GitHub repository with the task code so that readers can inspect all the details of the task and try it themselves.

3. I would recommend showing some power statistics for the results, which could provide more credibility to the findings and ensure the study's conclusions are supported by robust data analysis.

Response: This is a great point. We added the following text to the Methods section to quantify our experimental power.

With 40 trials per number modality, we were powered at 79.97% to detect a large effect size of 0.4 between the stimulus and inter-trial interval conditions.

---

## [Decision Letter · Decision Letter 1]

16 Sep 2024

PONE-D-24-19042R1Investigating the Triple Code Model in Numerical Cognition Using Stereotactic ElectroencephalographyPLOS ONE

Dear Dr. Lopez Ramos,

Thank you for submitting your manuscript to PLOS ONE. After careful consideration, we feel that it has merit but does not fully meet PLOS ONE’s publication criteria as it currently stands. Therefore, we invite you to submit a revised version of the manuscript that addresses the points raised during the review process.

We look forward to receiving your revised manuscript.

Kind regards,

Kiyoshi Nakahara, PhD

Academic Editor

PLOS ONE

Journal Requirements:

Reviewers' comments:

Reviewer's Responses to Questions

**Comments to the Author**

1. If the authors have adequately addressed your comments raised in a previous round of review and you feel that this manuscript is now acceptable for publication, you may indicate that here to bypass the “Comments to the Author” section, enter your conflict of interest statement in the “Confidential to Editor” section, and submit your "Accept" recommendation.

Reviewer #1: All comments have been addressed

2. Is the manuscript technically sound, and do the data support the conclusions?

Reviewer #1: Yes

3. Has the statistical analysis been performed appropriately and rigorously? 

Reviewer #1: Yes

4. Have the authors made all data underlying the findings in their manuscript fully available?

Reviewer #1: Yes

5. Is the manuscript presented in an intelligible fashion and written in standard English?

Reviewer #1: Yes

6. Review Comments to the Author

Reviewer #1: I have reviewed the revised manuscript and am pleased to see that the authors have thoroughly addressed all the concerns raised in the previous review. The revisions have significantly improved the quality and clarity of the paper. Given these improvements, I believe the manuscript is now almost suitable for publication in PLoS One. However, there are minor concerns that should be addressed before publication.

1. The caption of Figure 2 is still confusing, and I highly recommend improving its clarity. In particular, the explanation for panel c is unclear regarding which sub-panel corresponds to which description. For example, if I understand correctly, "(1)" and "(2)" are placed at the end of the explanations, "(4)" and "(5)" are at the top, and "(3)" is in the middle. Please ensure the caption accurately and clearly corresponds to the figure.

2. It appears that Figure 3 was significantly updated, but the caption was not. The caption mentions “a null binomial distribution (gray line),” which I do not find in the current figure. Instead, there are black vertical lines that are not explained in the caption. Additionally, it is still difficult to understand how the “null distribution” was computed based solely on the manuscript. Including the explanation provided in the response to the reviewers within the main text would be beneficial. Also, the closing parenthesis on line 206 seems to be a typo since there is no corresponding opening parenthesis..

3. The authors mentioned in the response to reviewers that they adopted a 70/30% training-test split, which yielded similar results to six-fold cross-validation. However, the revised main text states, “Six-fold cross-validation was used,” and there is no mention of the “70/30% training-test split” in the current manuscript. Please clarify which method was actually used in the analysis.

7. PLOS authors have the option to publish the peer review history of their article (what does this mean?). If published, this will include your full peer review and any attached files.

Reviewer #1: No

---

## [Author Response · Author response to Decision Letter 1]

14 Oct 2024

Thank you to both of the reviewers for your time. Please see below how we addressed the last few comments:

Reviewer #1:

1. The caption of Figure 2 is still confusing, and I highly recommend improving its clarity. In particular, the explanation for panel c is unclear regarding which sub-panel corresponds to which description. For example, if I understand correctly, "(1)" and "(2)" are placed at the end of the explanations, "(4)" and "(5)" are at the top, and "(3)" is in the middle. Please ensure the caption accurately and clearly corresponds to the figure.

The caption is reorganized so that all the letters references and number sub-references proceed their description. That we missed editing this on our final pass, thank you for catching it.

2. It appears that Figure 3 was significantly updated, but the caption was not. The caption mentions “a null binomial distribution (gray line),” which I do not find in the current figure. Instead, there are black vertical lines that are not explained in the caption. Additionally, it is still difficult to understand how the “null distribution” was computed based solely on the manuscript. Including the explanation provided in the response to the reviewers within the main text would be beneficial. Also, the closing parenthesis on line 206 seems to be a typo since there is no corresponding opening parenthesis.

The binomial distribution 99th percentile was remove for clarity since it is represented by the red-blue color divide of the plot but we missed removing the reference to the gray line, thank you for pointing that out, it would have been quite confusing! The text explaining this in the Response to Reviewers is now in the text and the parentheses typo has been fixed.

3. The authors mentioned in the response to reviewers that they adopted a 70/30% training-test split, which yielded similar results to six-fold cross-validation. However, the revised main text states, “Six-fold cross-validation was used,” and there is no mention of the “70/30% training-test split” in the current manuscript. Please clarify which method was actually used in the analysis.

Thank you to the reviewer for the excellent suggestion of using 6-fold cross-validation. We ended up presenting only the cross-validated results for clarity and have removed all references to the original 70:30 split. The results are likely more statistically robust cross-validated and so we decided that those were the results that should be presented (although they were quite similar).

---

## [Editor Report · Decision Letter 2]

21 Oct 2024

Investigating the Triple Code Model in Numerical Cognition Using Stereotactic Electroencephalography

PONE-D-24-19042R2

Dear Dr. Lopez Ramos,

We’re pleased to inform you that your manuscript has been judged scientifically suitable for publication and will be formally accepted for publication once it meets all outstanding technical requirements.

Kind regards,

Kiyoshi Nakahara, PhD

Academic Editor

PLOS ONE
---

## [Editor Report · Acceptance letter]

20 Nov 2024

PONE-D-24-19042R2 

PLOS ONE

Dear Dr. Lopez Ramos, 

I'm pleased to inform you that your manuscript has been deemed suitable for publication in PLOS ONE. Congratulations! Your manuscript is now being handed over to our production team.

Kind regards, 

on behalf of

Dr. Kiyoshi Nakahara 

Academic Editor

PLOS ONE